# Complete separation of benzene-cyclohexene-cyclohexane mixtures *via* temperature-dependent molecular sieving by a flexible chain-like coordination polymer

Feng Xie [1,5], Lihang Chen[2,5], Eder Moisés Cedeño Morales[3], Saif Ullah [4], Yiwen Fu[2], Timo Thonhauser [4], Kui Tan[3] ✉, Zongbi Bao [2] ✉ & Jing Li [1] ✉

The separation and purification of C6 cyclic hydrocarbons (benzene, cyclohexene, cyclohexane) represent a critically important but energy intensive process. Developing adsorptive separation technique to replace thermally driven distillation processes holds great promise to significantly reduce energy consumption. Here we report a flexible one-dimensional coordination polymer as an efficient adsorbent to discriminate ternary C6 cyclic hydrocarbons via an ideal molecular sieving mechanism. The compound undergoes fully reversible structural transformation associated with removal/re-coordination of water molecules and between activated and hydrocarbon-loaded forms. It exhibits distinct temperature- and adsorbate-dependent adsorption behavior which facilitates the complete separation of benzene, cyclohexene and cyclohexane from their binary and ternary mixtures, with the record-high uptake ratios for $C_6H_6/C_6H_{12}$ and $C_6H_{10}/C_6H_{12}$ in vapor phase and highest binary and ternary selectivities in liquid phase. In situ infrared spectroscopic analysis and ab initio calculations provide insight into the host-guest interactions and their effect on the preferential adsorption and structural transformation.

Hydrogenation of benzene ($C_6H_6$) to produce cyclohexene ($C_6H_{10}$) or cyclohexane ($C_6H_{12}$) is an industrially important petrochemical process[1,2]. Both cyclohexene and cyclohexane are indispensable raw chemicals widely used as feedstocks of cyclohexanol and cyclohexanone, which are eventually converted into adipic acid and caprolactam for nylon[3,4]. Currently, cyclohexene and cyclohexane are obtained primarily from catalytic (partial or full) hydrogenation of the benzene in ternary mixtures[5]. Simple distillation method fails to separate them because these triplets afford azeotropic mixtures with different compositions[6]. The difficulty largely arises from their close physical properties, including boiling points, molecular geometries, and relative volatilities[7,8]. The dominant industrial method for large-scale separation of benzene-cyclohexene-cyclohexane mixtures is based on specialized distillation processes *viz*, multi-column extraction and azeotropic distillation using dimethylacetamide and n-methyl-2-pyrrolidone as extractive agents[9–11]. Unfortunately, such agents suffer from low selectivity and poor solubility. As a result, multiple extraction cycles are required to achieve targeted purity

[1]Department of Chemistry and Chemical Biology, Rutgers University, 123 Bevier Road, Piscataway, NJ 08854, USA. [2]Key Laboratory of Biomass Chemical Engineering of Ministry of Education, College of Chemical and Biological Engineering, Zhejiang University, Hangzhou 310027, P. R. China. [3]Department of Chemistry, University of North Texas, 1155 Union Cir, Denton, TX 76203, USA. [4]Department of Physics and Center for Functional Materials, Wake Forest University, 1834 Wake Forest Road, Winston-Salem, NC 27109, USA. [5]These authors contributed equally: Feng Xie, Lihang Chen. ✉e-mail: kui.tan@unt.edu; baozb@zju.edu.cn; jingli@rutgers.edu

(>99%) at the cost of high equipment expense and extensive energy consumption[12–14]. To address these pitfalls, adsorptive separation techniques based on tailored nano-space of porous adsorbents are considered as an alternative and desirable solution for energy- and cost-efficient separation of benzene-cyclohexene-cyclohexane mixtures[15,16].

The emergence of microporous materials, particularly metal-organic frameworks (MOFs) or porous coordination polymers (PCPs), has stimulated strong interest in their applications for gas/vapor separation and purification[17–21]. The remarkable structural tunability and pore functionalizability of these materials have led to superior, sometimes unexpected, performance as adsorbents for a number of important gas/vapor separations[8,22]. However, to this date, there have been very few studies on selective separation of binary mixtures of C6 cyclic hydrocarbons, and no work has been reported on adsorption-based separations of benzene-cyclohexene-cyclohexane ternary mixtures[12,23–26]. Given the unique feature of dynamically flexible MOFs/PCPs and especially flexibility-related temperature-dependent adsorption phenomena[27–29], we have recently begun to explore the possibility of using them for efficient separation of C6 cyclic hydrocarbons. This study represents the example of complete separation of ternary mixture of C6 cyclic hydrocarbons via temperature-dependent molecule-sieving mechanism.

Temperature-dependent adsorption behavior induced by large hydrocarbons has been well documented for flexible porous MOFs and PCPs[28,30–36]. The framework typically undergoes breathing or swelling when exposed to various stimulus molecules at different temperature and pressure. Within a given temperature range, adsorbate molecules that interact most strongly with the adsorbent are capable of opening up the pore space and thus be adsorbed at all temperatures within the range (e.g., A in Fig. 1). On the contrary, molecules that interact most weakly with the adsorbent cannot open the pore space and will have negligible adsorption over the entire temperature range (e.g., C in Fig. 1). Molecules that interact less strongly than A but more strongly than C are likely to be excluded at higher temperature where interactions are not sufficiently high to open the pore open but adsorbed at lower temperature end (e.g., B in Fig. 1). This behavior allows full separation of the gas/vapor mixtures by the most preferred molecular sieving mechanism at different temperatures.

Here we demonstrate a manganese-based, highly-flexible one-dimensional (1D) coordination polymer, Mn-DHBQ (Mn(DHBQ)(H$_2$O)$_2$, H$_2$DHBQ = 2,5-dihydroxy-1,4-benzoquinone) is capable of discrimination of C$_6$H$_6$/C$_6$H$_{10}$/C$_6$H$_{12}$ mixtures in a specific temperature range. The overall structure of Mn-DHBQ is a hydrogen-bonded three-dimensional (3D) network composed of 1D Mn(DHBQ)(H$_2$O)$_2$ chains interconnected by hydrogen bonds. Upon activation, the network loses terminal water molecules and undergoes structure transformation to stacked 1D chains of Mn(DHBQ) which exhibits temperature- and adsorbate-dependent adsorption behavior toward three C6 cyclic hydrocarbons. This adsorbent demonstrates superior performance for molecular sieving based single-step separation of the C6 cyclic ternary mixtures with excellent selectivity and product purity.

## Results and discussion
Mn-DHBQ was synthesized by reaction of manganese acetate tetrahydrate and 2,5-dihydroxy-1,4-benzoquinone in aqueous solution at room temperature overnight[31,37,38]. A view of its crystal structure shows that each Mn(II) ion is equatorially coordinated to four oxygen atoms of two DHBQ ligands, and axially to two water molecules, giving rise to a strip-like 1D chain (Fig. 2a). Hydrogen bonds are formed between coordinated water and oxygen atoms of DHBQ of the adjacent chains resulting in a hydrogen-bonded 3D network with an OH···O distance of 1.80 Å (Fig. 2b, c). The network exhibits reversible structural change accompanied with the coordination/removal of water molecules (Supplementary Fig. 1).

The reversible structural changes accompanied by water removal and coordination were confirmed by powder X-ray diffraction (PXRD) analysis (Supplementary Fig. 2). The PXRD pattern of the activated sample (denoted as Mn(DHBQ)) is different from that of the as-made sample (denoted as Mn(DHBQ)(H$_2$O)$_2$), indicating a structure change during the activation process. The PXRD pattern taken after rehydration matches well with that of the as-made sample, demonstrating a full structural reversibility between the as-made and activated compound. The thermogravimetric analysis (TGA) confirmed that the weight loss up to 120 °C corresponds to two equivalents of water molecules (Supplementary Fig. 3). Moreover, the porosity and accessible pore volume of the activated sample were determined based on N$_2$ sorption data collected at 77 K. An apparent Brunauer-Emmett-Teller (BET) surface area was estimated to be ~380 m$^2$ g$^{-1}$ with an average pore size of ~5.5 Å. In contrast, negligible surface area was obtained for the as-made sample, verifying its nonporous nature. These results confirm that accessible pore is generated by removing coordinated water molecules (Supplementary Figs. 4 & 5). The pore channel and surface of the Mn-DHBQ framework with and without water molecules are presented in Fig. 2d and e, respectively.

While the as-made Mn(DHBQ)(H$_2$O)$_2$ shows no adsorption of the three C6 cyclic hydrocarbons (Supplementary Figs. 6 & 7), they can all be adsorbed by the activated Mn(DHBQ) sample, accompanied with a structural transformation (Supplementary Fig. 8). Although the pore size (~5.5 Å) of the activated structure appears somewhat too small to accommodate these molecules (C$_6$H$_6$: 6.628 × 7.337 × 3.277 Å$^3$; C$_6$H$_{10}$: 6.973 × 6.560 × 5.020 Å$^3$; C$_6$H$_{12}$: 7.168 × 6.580 × 4.982 Å$^3$)[39], the structure may be flexible enough to allow their entry. To evaluate the effect of structural flexibility of Mn(DHBQ) on its adsorption behavior, we carried out a series of adsorption experiments for C6 cyclic hydrocarbons at different temperatures. Single-component vapor static adsorption isotherms were collected at selected temperatures to investigate the intrinsic adsorption behavior and affinity of vapor molecules in Mn(DHBQ). The results clearly show that the adsorption affinity follows the trend of C$_6$H$_6$ > C$_6$H$_{10}$ > C$_6$H$_{12}$ (Supplementary Figs. 9–11). The C$_6$H$_6$ adsorption isotherms of Mn(DHBQ) show a distinct sharp step at relatively low pressures, indicating that it interacts most strongly and can be well accommodated in the channels. The uptake capacities are 2.90, 2.72, and 2.65 mmol g$^{-1}$ at 30, 60 and 90 °C, respectively. For C$_6$H$_{10}$, the adsorbed amounts are 2.80 and 2.01 mmol g$^{-1}$ at 30 and 60 °C, whereas at 90 °C, the uptake is negligible below 0.4 relative pressure (P/P$_{sat}$), followed by a slow increase at higher pressure, eventually reaching a total uptake of 0.97 mmol g$^{-1}$. On the other hand, the uptake of C$_6$H$_{12}$ is only 1.45 mmol g$^{-1}$ at 30 °C and negligible at 60 and 90 °C, with a loading of 0.08 and 0.02 mmol g$^{-1}$, respectively. Therefore, this molecule can be sieved out at both 60 and 90 °C. Notably, the uptake ratios, e.g., 130 for C$_6$H$_6$/C$_6$H$_{12}$ (90 °C) and 25 for C$_6$H$_{10}$/C$_6$H$_{12}$ (60 °C) are the highest among all porous adsorbents reported so far (Supplementary Table 6).

To evaluate the separation performance and feasibility under conditions similar to the industrial settings, single-component dynamic adsorption isotherms of C6 cyclic hydrocarbons were collected at 30, 60, and 90 °C in a continuous vapor flow (Fig. 3a, b & Supplementary Figs. 12–15). C$_6$H$_{12}$ is essentially excluded at both 60 and 90 °C, while C$_6$H$_6$ is adsorbed strongly at all three temperatures with the uptake capacities of 2.55, 2.32 and 2.23 mmol g$^{-1}$, respectively. The extent of C$_6$H$_{10}$ adsorption depends on the temperature. At 60 °C, about 1.72 mmol g$^{-1}$ is adsorbed, while the uptake at 90 °C is small, 0.43 mmol g$^{-1}$. Notably, the dynamic adsorption isotherms of C$_6$H$_6$ at both temperatures exhibit steep slopes, reaching adsorption saturation at very low pressure with fast kinetics, indicative of strong interactions with the Mn(DHBQ) framework (Supplementary Figs. 16–18). Comparison of the slopes in the linear region of the adsorption kinetics profile, it is clear that adsorption kinetics follows the trend C$_6$H$_6$ > C$_6$H$_{10}$ > C$_6$H$_{12}$ at all three temperatures. These results are

consistent with those from static adsorption experiments and suggest that $C_6H_{12}$ can be effectively sieved out at 60 and 90 °C, and $C_6H_{10}$ can be largely excluded at 90 °C.

The heat flows were measured at various temperatures (30, 60, and 90 °C) using a differential scanning calorimeter (DSC). The binding enthalpies (ΔH) of $C_6H_6$, $C_6H_{10}$, and $C_6H_{12}$ at 30 °C were estimated to be 76.3, 54.8, and 27.9 kJ mol⁻¹ respectively (Fig. 3c). At

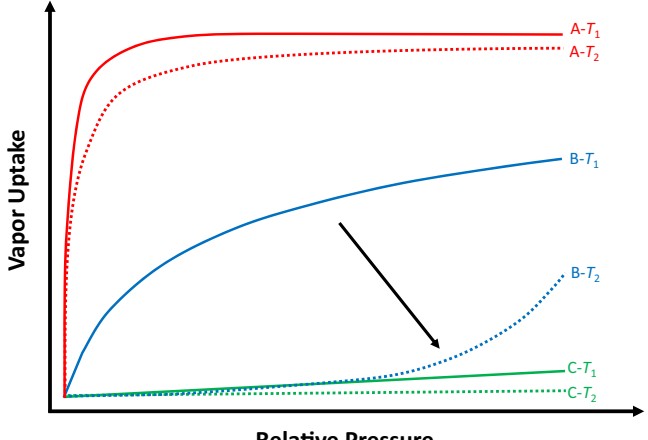

**Fig. 1 | The temperature-dependent vapor adsorption.** An illustration of temperature-dependent adsorption behavior observed for some vapor adsorbates (A, B and C) in flexible adsorbents. The trend of sorbent-sorbate interactions: A (red) > B (blue) > C (green); $T_1$ (solid lines) <$T_2$ (dotted lines). For the same type of adsorbates, the molecular size scale is typically A < B < C. The black arrow emphasizes the distinctly different adsorption behavior of molecule B at two temperatures. Such a behavior can be used to separate a ternary mixture of A/B/C at different temperatures.

60 and 90 °C, the values are 46.0 and 43.6 kJ mol⁻¹ for $C_6H_6$, 37.2 and 22.9 kJ mol⁻¹ for $C_6H_{10}$, and 13.0 and 11.5 kJ mol⁻¹ for $C_6H_{12}$ (Fig. 3d, e). These values are fully consistent with the observed temperature-dependent adsorption behavior of Mn(DHBQ). For the adsorbate molecules to enter the pore, certain energy is required to overcome the barrier (see calculated energy barriers in Supplementary Fig. 75) to stretch the pore open. With high ΔH values (43.6-76.3 kJ mol⁻¹), $C_6H_6$ is able to expand the interchain space and enter the pore at all three temperatures. However, the small ΔH values (22.9 kJ mol⁻¹ for $C_6H_{10}$ at 90 C, and 13.0/11.5 kJ mol⁻¹ for $C_6H_{12}$ at 60/90 °C) are insufficient to expand the pore space and to allow the molecules to fully enter. Additionally, the thermogravimetric analysis (TGA) of hydrocarbon-loaded Mn(DHBQ) samples also confirmed the order of interactions ($C_6H_6$ > $C_6H_{10}$ > $C_6H_{12}$) (Supplementary Fig. 19).

The results from single-component adsorption measurements suggest that separation of the three C6 cyclic hydrocarbons by molecular sieving mechanism may be achieved at different temperatures. To validate this experimentally, we conducted dynamic breakthrough experiments in fixed-bed columns. The separation performance of Mn(DHBQ) was evaluated for the binary and ternary mixtures of the three C6 cyclic hydrocarbons using one or two columns set at different temperatures. The results from breakthrough experiments are in excellent agreement with the trend of adsorption strength determined from the single-component vapor adsorption isotherms (Supplementary Figs. 20–36). At 90 °C, both $C_6H_{10}$ and $C_6H_{12}$ eluted out from the $C_6H_6$/$C_6H_{10}$ and $C_6H_6$/$C_6H_{12}$ binary mixtures shortly after the experiment started, whereas $C_6H_6$ did not break out until 140 and 70 mins later (Fig. 3f & Supplementary Fig. 25). As depicted in Supplementary Figs. 38 and 39, 0.72 and 0.96 mol kg⁻¹ of $C_6H_{12}$ and $C_6H_{10}$ were obtained, respectively, at this temperature, with a desirable ACS grade of over 99% purity. On the other hand, $C_6H_{12}$ eluted out immediately from the $C_6H_{10}$/$C_6H_{12}$ binary mixture at 30 °C while $C_6H_{10}$ broke out after ~120 mins, allowing a complete separation of $C_6H_{10}$ (Fig. 3g). This yielded 0.82 mol kg⁻¹ of $C_6H_{12}$ with a desirable ACS grade of over 99%

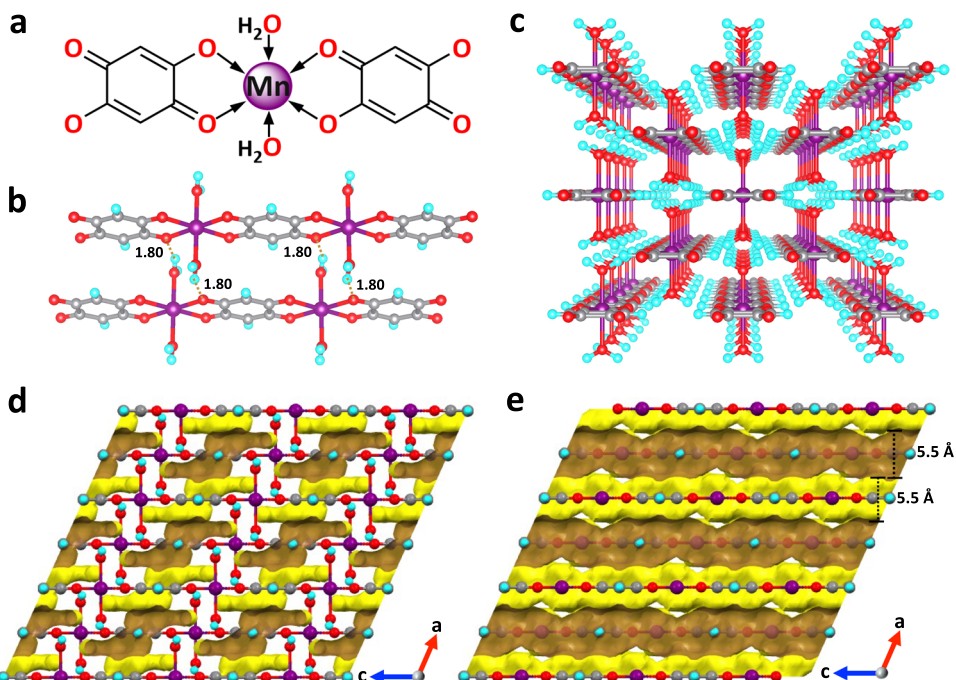

**Fig. 2 | Crystal and pore structure of Mn-DHBQ. a** The coordination sphere of Mn. **b** Interchain hydrogen bonds of 1.80 Å between the adjacent 1D chains (Mn, purple; O, red; C, grey; H, cyan). **c** Crystal structure of the as-made Mn(DHBQ)(H₂O)₂ viewed along the *c*-axis. **d** The isolated and inaccessible pore space (areas in brown and yellow colors) in the as-made Mn(DHBQ)(H₂O)₂ framework with coordinated water molecules. **e** The accessible pore channels in the Mn(DHBQ) framework if coordinated water molecules are removed from the original crystal structure. The pore channels are highlighted in brown and yellow colors with a size of ~5.5 Å (calculated from H-K model of the N₂ adsorption isotherm at 77 K).

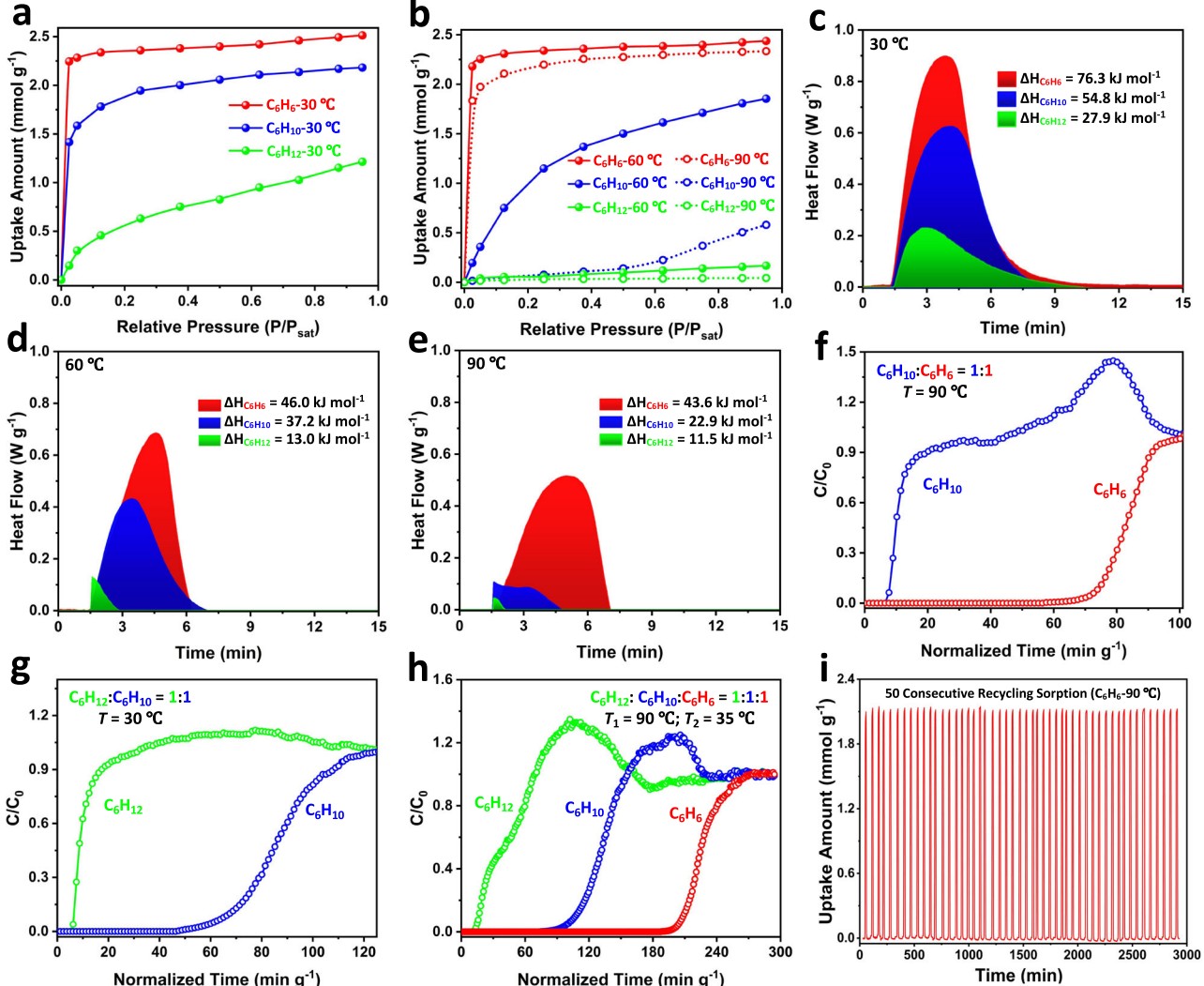

**Fig. 3 | Adsorption and separation of benzene-cyclohexene-cyclohexane on Mn-DHBQ. a** Dynamic vapor adsorption isotherms of C6 cyclic hydrocarbons on Mn(DHBQ) at 30 °C, (**b**) 60 and 90 °C. **c** Heat flows and ΔH values (<0) for the adsorption of C6 cyclic hydrocarbons on Mn(DHBQ) obtained from independent thermogravimetric differential scanning calorimeter (TG-DSC) measurements at 30 °C, (**d**) 60 °C, and (**e**) 90 °C. **f** Breakthrough curves of an equimolar binary $C_6H_6$/$C_6H_{10}$ mixture on Mn(DHBQ) pellets packed in the single column at 90 °C. **g** Breakthrough curves of an equimolar binary $C_6H_{10}$/$C_6H_{12}$ mixture on Mn(DHBQ) pellets packed in the single column at 30 °C. **h** Breakthrough curves of an equimolar ternary mixture of C6 cyclic hydrocarbons on Mn(DHBQ) pellets packed in two connected columns at 90 and 35 °C, respectively. **i** Adsorption-desorption recyclability tests of $C_6H_6$ on Mn(DHBQ) for 50 consecutive sorption cycles at 90 °C.

purity (Supplementary Fig. 40). Thus, a one-step separation of $C_6H_6$/$C_6H_{10}$/$C_6H_{12}$ ternary mixture could be achieved by using multicolumn separation technique at different temperatures.

The one-step separation of ternary equimolar $C_6H_6$/$C_6H_{10}$/$C_6H_{12}$ mixtures was performed using two fixed-bed columns set at different temperatures (Supplementary Fig. 41). At 90 °C, $C_6H_6$ was adsorbed in the first column and fully separated from the ternary mixture, followed by a complete separation of $C_6H_{10}$ from the remaining binary mixture of $C_6H_{10}$ and $C_6H_{12}$ in the second column at 35 °C (Fig. 3h & Supplementary Fig. 42). Moreover, multiple recycling sorption experiments demonstrated the excellent recyclability of Mn(DHBQ). The uptake amount, surface area, and crystallinity remained unchanged after 50 consecutive adsorption-desorption cycles at 90 and 60 °C for $C_6H_6$ and $C_6H_{10}$ (Fig. 3i & Supplementary Figs. 43–46). In addition, Mn-DHBQ exhibits excellent stability toward acid, base, water, and heat over long period of time, as confirmed by the porosity and PXRD measurements after various treatments (Supplementary Fig. 47). In the reproducibility tests, we carried out a set of six parallel reactions under identical conditions (Supplementary Fig. 48). All reactions produced

high-quality samples with quantitative yields. Their crystallinity, BET surface area, and temperature-dependent adsorption behavior are essentially the same and closely resemble those of the previously synthesized samples (Supplementary Figs. 49–51), confirming the excellent reproducibility of the synthesis and suitability of Mn(DHBQ) for potential industrial applications. To further verify its performance enhancement, we also conducted control experiments using an industrial standard adsorbent material, ZSM-5. A close comparison of the results from adsorption and breakthrough experiments (Supplementary Figs. 52–54) highlights the superior performance of Mn(DHBQ) in the separation of C6 cyclic hydrocarbons.

To quantify the discrimination selectivity of Mn(DHBQ) for the ternary mixtures of $C_6H_6$/$C_6H_{10}$/$C_6H_{12}$, competitive adsorption measurements were performed in both vapor and liquid phases. The data were analyzed by proton nuclear magnetic resonance ($^1$H-NMR) and gas chromatography (GC). For the equimolar binary vapor mixtures (Supplementary Figs. 55–57, 59), the relative adsorption selectivities of Mn(DHBQ) are 37.3/1 and 172.3/1 for $C_6H_6$/$C_6H_{10}$ and $C_6H_6$/$C_6H_{12}$ at 90 °C (Supplementary Figs. 61 & 62), respectively, and 9.6/1 for $C_6H_{10}$/

$C_6H_{12}$ at 60 °C (Supplementary Fig. 63). Additionally, a high selectivity of 107.8/3.1/1 for $C_6H_6$ over the other two C6 cyclic species was observed for an equimolar ternary vapor mixture of $C_6H_6/C_6H_{10}/C_6H_{12}$ at 90 °C (Supplementary Figs. 58, 60, 64). Moreover, static solid-liquid extraction experiments were carried out at different temperatures to evaluate separation performance in the liquid phase. The results confirmed the distinct adsorption affinity and selectivity on Mn(DHBQ) in the order of $C_6H_6 \gg C_6H_{10} \gg C_6H_{12}$, especially at higher temperature (Supplementary Figs. 65 & 66). At 60 and 30 °C, its selectivities for the ternary mixture of $C_6H_6/C_6H_{10}/C_6H_{12}$ based on GC analysis are 642.7/11.3/1 and 387.1/7.8/1, respectively (Supplementary Fig. 67), representing the highest values reported so far for liquid-phase separation by porous adsorbents. These results indicate that Mn(DHBQ) is a very promising adsorbent for the separation of $C_6H_6/C_6H_{10}/C_6H_{12}$ ternary mixtures.

To unravel the relationship between the structural flexibility and the observed temperature- and adsorbate-dependent adsorption behavior of Mn-DHBQ, we performed a structural analysis of hydrocarbon-adsorbed Mn(DHBQ) samples employing ex situ powder X-ray diffraction (ex situ PXRD) method and in situ infrared spectroscopy (in situ IR). While the activated Mn(DHBQ) sample remained crystalline, the significant broadening of the PXRD peaks suggests a high level of local disorders due to the removal of coordinated water molecules, which largely diminished interchain interactions, giving rise to a more flexible structure. As the adsorption took place on the activated sample, it underwent a structure change/transformation to accommodate the guest molecules while retaining its periodic order. Whether the molecule can be adsorbed depends on the relative binding strength between the adsorbate and adsorbent. In the case of $C_6H_6$, its strong interaction with Mn(DHBQ) enables opening of the constrained pore space even at higher temperatures (e.g., 90 °C), therefore, adsorption can occur at all three temperatures (30, 60 and 90 °C) and in all cases, the resultant structures of $C_6H_6$-loaded samples at these temperatures are identical (Fig. 4a & Supplementary Fig. 68).

For $C_6H_{10}$, as it interacts with Mn(DHBQ) less strongly, the constrained pore space can be pushed open only at lower temperatures (30 and 60 °C), at which adsorption and corresponding structure transformation were observed (Fig. 4a & Supplementary Fig. 69). $C_6H_{12}$, on the other hand, has the least interaction with Mn(DHBQ) and can only be partially adsorbed at 30 °C with an incomplete structure change (Fig. 4a & Supplementary Fig. 70). This temperature- and adsorbate-dependent behavior forms the basis of the molecular sieving governed separation of the ternary mixture: the individual C6 cyclic species can be sieved out at a suitable temperature (Fig. 3h & Supplementary Fig. 41).

The results from in situ infrared (IR) spectroscopy experiments provided further details for a microscopic understanding of host-guest interactions and structural transformation. The spectrum of the as-made Mn(DHBQ)(H$_2$O)$_2$ was measured first, which is dominated by the absorption bands associated with vibrations of deprotonated DHBQ linker (Fig. 4b). The previous studies that extensively investigated the infrared spectra of DHBQ metal-complex compounds[40–44], together with the spectrum of the free ligand of DHBQ acid (Fig. 4b)[45], make it possible to establish an unambiguous assignment of the observed bands in the as-made Mn(DHBQ)(H$_2$O)$_2$ compound, which is a crucial step toward determining the subtle structural changes occurring in specific bonds upon removal of water molecules and further loading of guest hydrocarbons. Supplementary Table 4 & Supplementary Note 1 summarize the detailed assignment of observed bands in Fig. 4b, c.

The as-made Mn(DHBQ)(H$_2$O)$_2$ sample was then activated in situ to 150 °C. Figure 4b shows that the interchain hydrogen bonded water, typified by its downward (red-) shifted and broadened $\nu$(OH) band at 3313 cm$^{-1}$ and upward (blue-) shifted $\beta$(H$_2$O) band at 1640 cm$^{-1}$[46], starts to desorb with increasing the temperature above 60 °C. Along with the departure of water, marked changes occur to several DHBQ bands as shown in Fig. 4b & Supplementary Table 4. For instance, a sharp band becomes clearly observable at 1615 cm$^{-1}$, which is assigned to phenyl C = C stretching that occurs at ~1597 cm$^{-1}$ in the as-made form[43,45,47].

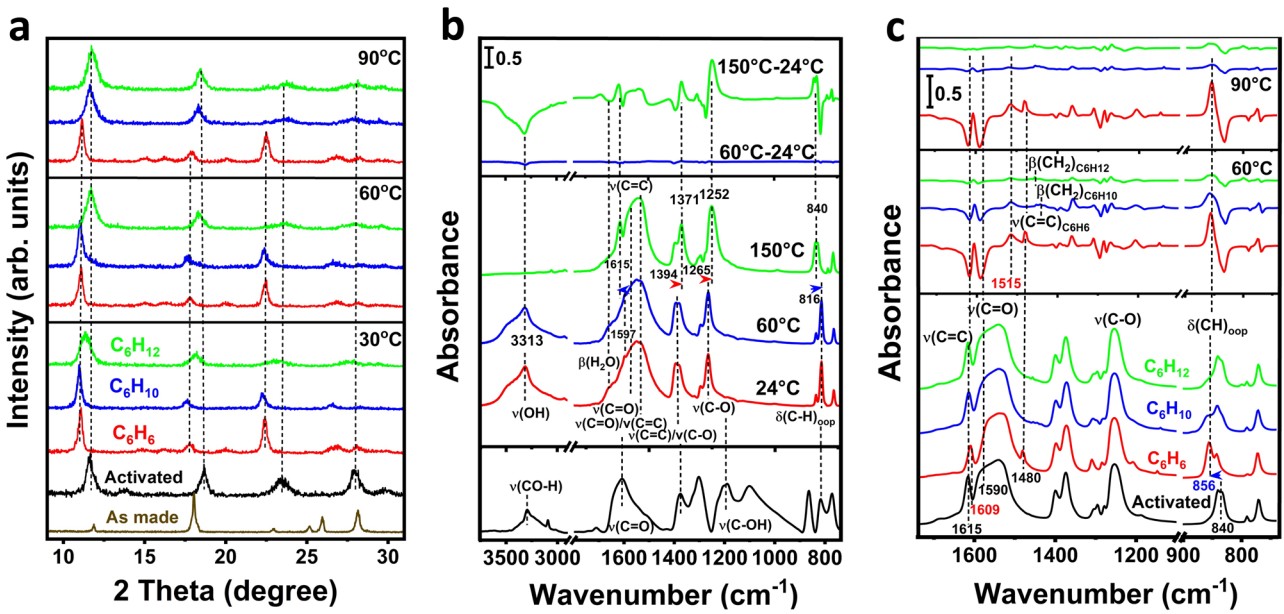

**Fig. 4 | Temperature-dependent structural transformation of Mn-DHBQ. a** Ex situ PXRD patterns of activated and hydrocarbon-adsorbed Mn(DHBQ) at 30, 60, and 90 °C, respectively. **b** In situ IR spectra of activating Mn-DHBQ upon heating under vacuum. Bottom panel: the spectrum of free DHBQ ligand in solid form; Middle panel: the spectra of Mn(DHBQ) at different temperatures; Top panel: difference spectra obtained by subtracting the spectrum taken at 24 °C from that at 60 and 150 °C to show the changes upon activation. **c** In situ IR spectra of loading

hydrocarbons (at ~50 Torr for ~8 min) into activated Mn(DHBQ). Bottom panel: IR spectra of activated and hydrocarbon-loaded Mn(DHBQ) samples at 60 °C; Middle and top panels: difference spectra obtained by subtracting the spectrum of the activated sample from those of hydrocarbon-loaded ones at 60 and 90 °C, respectively. Notation and acronym: $\nu$ = stretching, $\delta$ = deformation, $\beta$ = bending (scissoring), and oop = out of plane.

The blue-shift of ~18 cm⁻¹ indicates the hardening and/or shortening of the C = C bond. We have also noticed that mixed $\nu$(C = C)/$\nu$(C−O) and localized $\nu$(C−O) bands slightly red-shift to 1371 and 1252 cm⁻¹, respectively, which indicates that the C−O bond is released from hydrogen bonding with the interchain water. Furthermore, the $\delta$(CH)$_{oop}$ band in the phenyl ring at 816 cm⁻¹ shifts to a higher frequency at 840 cm⁻¹, suggesting a narrowing of the interspace of 1D chains that impedes out-of-plane CH bending vibration[48]. These informative bands were further tracked upon loading hydrocarbons to elucidate their interaction within Mn(DHBQ) and resulting structural changes. C₆H₆ vapor was loaded first at 60 °C and shows significant adsorption, typified by its characteristic bands including $\nu$(CH) at ~3100-3000 cm⁻¹, $\nu$(CC) at 1480 cm⁻¹ and $\omega$(CH) at 675 cm⁻¹ in the difference spectrum in Fig. 4c & Supplementary Fig. 71. The red-shift of $\nu$(CC) band by ~6 cm⁻¹ regarding the gas phase value at 1486 cm⁻¹ suggests that benzene is adsorbed at the exposed Mn(II) sites through Chatt-Dewar-Ducanson mechanism that involves $\sigma$-donation and $\pi$ back donation between benzene and Mn(II)[49,50]. Such interaction usually results in the weakening of the phenyl bond and hence a decrease in its frequency[48]. More remarkably, DHBQ bands as discussed above exhibit clear changes after adsorbing benzene: (1) a red shift of $\nu$(C = C) band, from 1615 to 1609 cm⁻¹; (2) a loss of feature at 1590 cm⁻¹ and gain at 1515 cm⁻¹ as indicated by the difference spectrum in Fig. 4c, pointing to the red shift of $\nu$(C = O) band; (3) blue shift of $\delta$(CH)$_{oop}$ band from 840 cm⁻¹ to 856 cm⁻¹. These observations shed important light on the origin of Mn-DHBQ structural changes. For instance, the red shift of $\nu$(C = O) and $\nu$(C = C) bands suggests the softening and/or elongating of those chemical bonds[51], consistent with the PXRD observation of Mn(DHBQ) structural expansion after adsorbing C₆H₆. Compared with the carbonyl group, phenolate band $\nu$(C-O) is less affected (Fig. 4c), indicating that C = O-M moiety is more flexible than C-O-M upon exposure to guest stimuli. The blue shift of $\delta$(CH)$_{oop}$ band implies that the inclusion of guest C₆H₆ further impedes C-H bending motion, thus raising its frequency. These findings are in good agreement with our computational prediction that benzene binds onto Mn(II) sites and establishes multipoint interactions with neighboring O and CH of DHBQ linker. The adsorbed C₆H₆ can be fully desorbed by evacuation under vacuum and the Mn-DHBQ structure recovers to its initial activated form after the removal of C₆H₆ (Supplementary Figs. 71 & 72). Reloading of C₆H₆ was conducted at 90 °C under identical conditions, giving rise to a similar result (Fig. 4c). After regenerating the sample (by annealing up to 120 °C under vacuum), we further loaded C₆H₁₀ vapor at 60 °C. The adsorption of C₆H₁₀ is characterized by its typical bands such as $\nu$(CH) at ~3000−2800 cm⁻¹, $\beta$(-CH₂) at 1445 cm⁻¹ (Supplementary Fig. 73), and accompanied by structural change of Mn(DHBQ), as indicated by the perturbed phonon modes including $\nu$(C = C), $\nu$(C = O), and $\delta$(CH)$_{oop}$. Compared with the C₆H₆-loaded sample, C₆H₁₀-loaded one exhibits weaker perturbations, e.g., $\nu$(C = C) of DHBQ at 1615 cm⁻¹ only red-shifts by ~2 cm⁻¹; partial growth of perturbed $\nu$(C = O) at 1515 cm⁻¹ and $\delta$(CH)$_{oop}$ at 856 cm⁻¹ is seen at 60 °C. After regenerating the sample, the reloading experiment

was conducted at 90 °C (Fig. 4c). In this case, only slight guest adsorption and host structural perturbations were observed, consistent with the isotherm results. Lastly, we exposed the regenerated sample to C₆H₁₂ at 60 and 90 °C, respectively. In this case, only minimal C₆H₁₂ was adsorbed, and compound skeleton modes show no appreciable changes.

To further investigate the guest-host interactions, ab initio calculations were performed to provide molecular level mechanistic insights into the types and extent of interactions between Mn(DHBQ) and C6 cyclic hydrocarbons (Fig. 5a–c). We started by loading C6 cyclic hydrocarbon into the interchain space of Mn(DHBQ) to determine the optimal binding sites. Based on the guest-host interaction, the strongest binding is found for C₆H₆ with a binding energy of 0.57 eV or 55.0 kJ mol⁻¹ (Supplementary Table 5). A slightly lower binding energy of 0.47 eV (45.3 kJ mol⁻¹) is found for C₆H₁₀. However, due to its larger size and somewhat different structure, the binding energy for C₆H₁₂ is only 0.16 eV or 15.5 kJ mol⁻¹. The corresponding diffusion energy barriers calculated for the three molecules are plotted in Supplementary Fig. 74. We find a quite low diffusion barrier of 0.08 eV for C₆H₆ compared to 0.14 eV for C₆H₁₀. Consequently, in addition to the thermodynamic effect, a kinetic effect is also at play between these hydrocarbons. A significantly higher barrier of 0.29 eV is calculated for C₆H₁₂, thus the separation of C₆H₆/C₆H₁₀ vs C₆H₁₂ is both thermodynamically and kinetically driven. Our ab initio calculation results are in perfect agreement with the experimental findings. We also estimated the energy required to stretch the interchain space to accommodate the three C6 molecules (Supplementary Fig. 75). The strains of the three molecules clearly follow the trend: C₆H₆ < C₆H₁₀ < C₆H₁₂. Assuming that adsorption follows standard thermodynamics, we calculated temperature-dependent isotherms using a simple Langmuir isotherm model and compared them with experimentally measured data. As shown in Supplementary Figs. 76−78, the temperature-dependence of the experimental isotherms deviate significantly from what standard thermodynamics – in the form of the Van't Hoff equation in Langmuir isotherms – would predict.

To gain further mechanistic understanding regarding the host-guest interaction, induced charge densities, i.e., the charge rearrangement upon bond formation was computed and the results show that the exposed C atoms in benzene interact quite strongly with the Mn of the framework from the adjacent chains (Supplementary Fig. 79). The C atoms in C₆H₁₀ are only semi-exposed, and only two adjacent exposed C atoms make strong bonds with the Mn (Supplementary Fig. 80). Apart from this main interaction site, a prominent interaction of C₆H₆ and C₆H₁₀ with the adjacent O and CH of the DHBQ linker was also observed. On the other hand, in C₆H₁₂ none of the C atoms are exposed, preventing a direct interaction with the Mn in the chains (Supplementary Fig. 81). All these results match well with the experimental isotherms and in situ IR data. The combined ex situ PXRD analysis, in situ IR spectroscopy study and ab initio calculations offer insightful information regarding detailed guest-host interactions and specific changes in the chemical bonds. The as-made structure shrinks

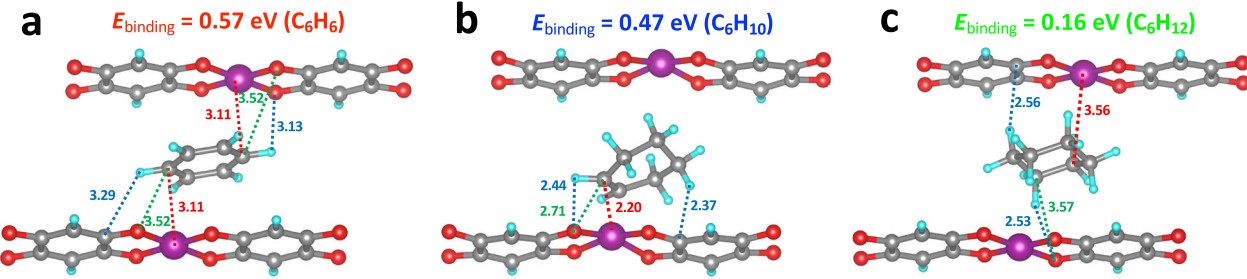

**Fig. 5 | Mechanism illustration of host-guest interactions in Mn-DHBQ.** The binding interactions between C6 cyclic hydrocarbons (C₆H₆ (**a**), C₆H₁₀ (**b**), and C₆H₁₂ (**c**)) and chains of Mn(DHBQ) as determined by ab initio calculations.

upon activation due to the removal of water molecules and expands upon exposure to hydrocarbons. The main interactions are with the Mn(II) and secondary interactions are with O and CH of the DHBQ linker. IR spectroscopic study shows that the M-O = C and phenyl ring C = C bonds exhibit a certain flexibility upon loading of guest molecules, as evidenced by the shift of their stretching bands, thus providing a path of pore expansion.

In conclusion, $C_6H_6$, a cyclic hydrocarbon, as well as its derivatives such as $C_6H_{10}$ and $C_6H_{12}$, are important feedstocks for the production of many polymers, plastics and fibers. These molecular species usually co-exist in the $C_6H_6$ hydrogenation process. Having nearly identical physical properties (e.g., boiling point and molecular size), the industrial separation of these molecules typically relies on azeotropic and extractive distillations, which require expensive equipment and intensive energy input. To address these issues, we have explored low-cost and energy-efficient adsorptive separation using a coordination polymer composed of 1D-Mn(DHBQ)(H$_2$O)$_2$ chains that are interconnected via hydrogen bonds to form a 3D H-bonded network. The compound features high structural flexibility and abundant open metal sites upon activation. Its unique temperature-dependent adsorption behavior allows, for the first time, selective adsorption of C6 cyclic hydrocarbons and fast separation of their ternary mixtures based on molecular sieving mechanism. The guest-induced structural rearrangement of the adsorbent and the origin of its preferential interactions with specific C6 cyclic hydrocarbons are elucidated and explained by ex situ powder X-ray diffraction, in situ infrared spectroscopy, and ab initio calculations. Additionally, the record-high uptake ratios of $C_6H_6$/$C_6H_{12}$ and $C_6H_{10}$/$C_6H_{12}$ in vapor phase, and the high selectivities for both binary and ternary mixtures in liquid phase are achieved. Furthermore, the facile synthesis, cost-effective ligand, excellent chemical/thermal stability and recyclability render Mn-DHBQ a very promising candidate for industrial implementation of highly efficient adsorptive separation of C6 cyclic hydrocarbons.

# Methods

## Chemicals
All chemicals and gases were used as received without any further purification. 2,5-dihydroxy-1,4-benzoquinone (H$_2$DHBQ, 98%), manganese acetate tetrahydrate (Mn(CH$_3$COO)$_2$·4H$_2$O, 99%) were purchased from Sigma-Aldrich Co. (USA). The nitrogen used for the adsorption isotherms record shows a purity of 99.999% from Paraxair Co. (USA). Benzene ($C_6H_6$, >99%), cyclohexene ($C_6H_{10}$, >99%), and cyclohexane ($C_6H_{12}$, >99%) were purchased from Sigma-Aldrich Co. (USA).

## Mn-DHBQ synthesis
Mn-DHBQ was prepared following the method previously reported with minor modifications and can be produced in 100 g scale. 2,5-dihydroxy-1,4-benzoquinone (H$_2$DHBQ, 1 mmol) was dissolved in deionized water (1 eq, 0.1 M). An ambient aqueous solution of manganese acetate tetrahydrate (1 eq, 0.2 M) was added to the stirring mixture at room temperature overnight. The resulting powder was filtered and activated at 150 °C under dynamic vacuum overnight for following various characterizations and adsorption measurements.

## Mn-DHBQ pellets preparation
For practical separation applications, scalable pellets were prepared in the home-made mold. Specifically, a certain amount of Mn-DHBQ powder samples was mixed uniformly with ~5 wt% of hydroxypropyl cellulose binder and an appropriate amount of deionized water to form a paste. The resulting paste was partially dried at ambient condition and knead for the pellets. The pellet diameter was controlled at 1–2 mm and the single weight was about 5–6 mg. The as-made pellets were further activated at 150 °C under dynamic nitrogen environment for subsequent column breakthrough tests.

## Thermogravimetric analysis
Thermal stability of Mn-DHBQ was performed on the TA Instrument Q5000IR thermal gravimetric analyzer. About 10 mg of powder sample was loaded onto a platinum pan and heated under nitrogen flux with a heating rate of 5 K min$^{-1}$ from room temperature to 700 °C.

## Powder X-ray diffraction (PXRD)
PXRD measurements were performed on a Rigaku Ultima-IV automated diffraction system (Rigaku Corp., Japan) using Cu-Kα radiation (λ = 1.5406 Å) in the 2 Theta range of 3-40° with a scan rate of 3° min$^{-1}$. The operating powder was 40 kV/44 mA.

## Gas sorption & porosity analysis
Surface area and pore size distribution were measured by nitrogen adsorption-desorption isotherms at 77 K for different treatment Mn-DHBQ, including as-made Mn-DHBQ and activated Mn-DHBQ powder samples. The adsorption-desorption isotherms were obtained on a Micromeritics 3Flex adsorption analyzer (Micrometrics Instrument Corp., USA). About 100 mg powder samples were activated at 150 °C under dynamic vacuum overnight before surface area and pore size tests.

## Single-component static vapor adsorption isotherms
The samples of Mn-DHBQ were activated at 150 °C overnight under dynamic vacuum prior to single-component hydrocarbon vapor adsorption measurements. The single-component isotherms of $C_6H_6$, $C_6H_{10}$, and $C_6H_{12}$ were measured at 30, 60, 90 °C on a Micromeritics 3Flex adsorption analyzer equipped with a vapor dosing bottle. Each sample tube was subsequently immersed in a temperature-controlled heating mantle that surrounded most of the sample tube. The manifold of the instrument itself, including the vapor dosing bottle was heated to 25 °C and kept at this temperature for all these single-component C6 cyclic hydrocarbon measurements. The saturated pressure (P$_{sat}$) was defined as the vapor pressure of each hydrocarbon at 25 °C: $C_6H_6$ (12.7 kPa), $C_6H_{10}$ (12.0 kPa), $C_6H_{12}$ (13.0 kPa).

## Single-component dynamic vapor adsorption isotherms
Adsorption isotherms can be determined experimentally by using either static or dynamic measuring methods. Compared with the static adsorption which takes place in the enclosed chamber, dynamic adsorption occurs in a flow system, very similar to the industry separation processes. Dynamic vapor adsorption isotherms measurements were performed on a homemade gravimetric adsorption unit modified from a TGA Q50 thermogravimetric analyzer (TA Instruments). Pure nitrogen was used as a carrier gas passing through a bubbler filled with single-component liquid hydrocarbon. The partial pressures of hydrocarbon were controlled by adjusting the blend ratio of pure nitrogen and mixed nitrogen with saturated hydrocarbon vapor. The absorbed amount was monitored by the weight change of sample over the experiment period in the TGA computer system. About 20 mg Mn-DHBQ powder sample were activated at 150 °C under nitrogen flow for 1 hr to remove any residual water molecules. The temperature was then cooled down to the adsorption temperature (30, 45, 60, and 90 °C), and another nitrogen flow passing through a hydrocarbon bubbler at room temperature was mixed with the pure nitrogen stream. The mixed gases stream (with the total flow rate of 40 mL min$^{-1}$) was finally introduced to the adsorption chamber where the temperature can be controlled with the heating furnace and circulating water. The recorded weights of adsorption were converted into transient normalized uptake, reflecting the adsorption kinetics over time. Each hydrocarbon's adsorption was monitored for 15 minutes to ensure equilibrium with saturated capacity was reached. To assess the kinetic effects and variations among C6 cyclic hydrocarbons in Mn-DHBQ, we performed a brief fitting and calculated the slope of uptake amount over time from the onset of adsorption to saturation

equilibrium. The saturated pressure ($P_{sat}$) was defined as the vapor pressure of each hydrocarbon at 25 °C: $C_6H_6$ (12.7 kPa), $C_6H_{10}$ (12.0 kPa), $C_6H_{12}$ (13.0 kPa).

## Vapor-phase competitive adsorption analysis by nuclear magnetic resonance (NMR)

Adsorption selectivity analysis for the binary and ternary mixtures of C6 cyclic hydrocarbons was carried out using NMR at various adsorption temperatures. Typically, about 20 mg sample of as-made Mn-DHBQ was placed into the adsorption chamber and activated at 150 °C for 1 hr prior to hydrocarbon vapor introduction. The equimolar binary or ternary hydrocarbon mixture was then introduced by the nitrogen stream at the designed adsorption temperature. Specifically, the chamber was controlled at 60 °C to measure the adsorption capacity and separation selectivity of equimolar binary mixture of $C_6H_{10}/C_6H_{12}$ vapors. For the equimolar binary/ternary vapor mixtures of $C_6H_6/C_6H_{10}$, $C_6H_6/C_6H_{12}$, and $C_6H_6/C_6H_{10}/C_6H_{12}$, the adsorption chamber was operated at 90 °C. The relative pressure was set to $P/P_{sat} = 0.75$ for the hydrocarbon vapor related to total pressure ($P_{sat}$) of each adsorbate molecule. After achieving adsorption equilibrium with the specific hydrocarbon vapor, each sample was transferred into sealed glass tube. To completely exchange the adsorbed hydrocarbon vapor, 2 mL concentrated hydrochloric acid was added to each glass tube to decompose the Mn-DHBQ sample, and then 2 mL $CDCl_3$ was added to the mixture, allowing a complete exchange of the corresponding hydrocarbon by $CDCl_3$ after 24 hrs at room temperature. The lower organic layer containing C6 cyclic hydrocarbon was collected and analyzed by $^1H$ NMR. Notably, to verify the feasibility and make the peak comparison easier, the relevant peaks of equimolar binary or ternary of C6 cyclic hydrocarbons were collected first.

## Liquid-phase competitive adsorption analysis by gas chromatography (GC)

To accurately quantify the uptake of individual C6 cyclic hydrocarbon in the Mn-DHBQ, the hydrocarbon-loaded samples collected from the static solid-liquid extraction experiments were destroyed by concentrated hydrochloric acid (36 ~ 38 wt%) followed by $CH_2Cl_2$ extraction. The binary/ternary equimolar mixtures of C6 cyclic hydrocarbons were prepared by combining $C_6H_6$, $C_6H_{10}$ and/or $C_6H_{12}$ in appropriate quantities weighed accurately. In a typical experiment, about 50 mg of freshly activated Mn-DHBQ samples were soaked in 6 g (~7 mL) of a binary or ternary mixture and the samples were allowed to equilibrate at 30 and 60 °C, respectively. After 1 hr, the solid adsorbents were collected by filtering at the adsorption temperature and were allowed to dry on filter paper for 30 mins to remove C6 cyclic hydrocarbons adhered to the surface of samples, after which the hydrocarbon adsorbed Mn-DHBQ samples were decomposed by 2 mL concentrated hydrochloric acid. The destroyed Mn-DHBQ samples were then immersed in 2 mL $CH_2Cl_2$ overnight for enabling the hydrocarbon to be completely extracted by $CH_2Cl_2$. The bottom organic layer was collected for GC measurements. The peak areas of each hydrocarbon shown in chromatograms were used to calculate the selectivity. Adsorption selectivity, $S_{ij}$, is defined by the following equation:

$$S_{ij} = \frac{x_i/y_i}{x_j/y_j} \qquad (1)$$

where $x_i$, $x_j$ are the equilibrated adsorption capacity of component $i$ and $j$ in the adsorbed phase, respectively; and $y_i$, $y_j$ are the molar fraction of component $i$ and $j$ in the vapor phase or liquid phase. For equimolar mixtures, the selectivity can be simplified as

$$S_{ij} = \frac{x_i}{x_j} \qquad (2)$$

The ratio of $x_i/x_j$ can be derived from the integrated area ratio of individual C6 cyclic hydrocarbon species in $^1H$-NMR spectra or the peak areas in GC chromatograms.

## Recyclability tests

The recyclability tests of Mn-DHBQ were performed on TGA-Q50 gravimetric adsorption analyzer for 50 consecutive benzene and cyclohexene adsorption-desorption cycles at various temperatures. About 20 mg of Mn-DHBQ samples were activated at 150 °C under nitrogen flow for 1 hr before the experiments. After cooling down to the adsorption temperature (90 °C for $C_6H_6$, 60 °C for $C_6H_{10}$), another nitrogen flow passing through a liquid $C_6H_6$ or $C_6H_{10}$ bubbler was introduced to the adsorption chamber. After the adsorption reached equilibrium (-20 mins), the hydrocarbon-adsorbed samples were desorbed at 150 °C for 1 hr to reactivate the Mn-DHBQ sample for the next adsorption-desorption cycle.

## Reproducibility tests

The reproducibility of Mn-DHBQ was assessed by performing parallel experiments on six batches of samples, which were synthesized under the same synthetic conditions. In each reaction, 1 mmol $Mn(OAc)_2$ was mixed with 1 mmol $H_2DHBQ$ in a 20 mL aqueous solution, and the mixture was stirred at room temperature overnight. The resulting products were collected via centrifugation and drying under ambient conditions for subsequent characterizations. Assessment of sample crystallinity and porosity was carried out by PXRD and porosity (BET surface area and pore size distribution) analyses. In addition, temperature-dependent adsorption isotherms were collected for $C_6H_6$ using a TGA-Q50 gravimetric adsorption analyzer at different temperatures (60 and 90 °C).

## Multicomponent vapor breakthrough experiments

Multicomponent column breakthrough experiments were conducted with the activated Mn-DHBQ pellets (about 2 g) that were packed into a stainless steel HPLC column (L: 50 mm, I.D.: 10 mm). The packed pellets were activated at 150 °C under nitrogen flow for 1 hr before vapor breakthrough experiments. The binary or ternary mixture of $C_6H_6/C_6H_{10}/C_6H_{12}$ was loaded into a glass bubbler at room temperature, which is connected to the sample packed column. The flow of nitrogen was passed through the hydrocarbon bubbler (at a constant 25 °C by a thermostatic water bath) with a rate of 5 mL min$^{-1}$. The composition of the binary or ternary components in the bubbler was adjusted to achieve an equimolar mixture. The temperature of the sample packed column was controlled in an oil bath at 30, 35, 45, 60, and 90 °C, to evaluate the C6 cyclic hydrocarbon separation performance on the Mn-DHBQ pellet samples at various temperatures. The vapor at the outlet of the sample-packed column was monitored using the gas chromatograph (GC-2010 Pro SHIMADZU) with a flame ionization detector. The vapor mixture was separated by a capillary column (PEG-20M, Φ0.32 × 60 m) at 70 °C. Notably, the time caused by the void volume of the pipeline and the column have been deducted when processing the breakthrough data.

To calculate the component ratios of the mixture, the saturated pressures of $C_6H_6$, $C_6H_{10}$, and $C_6H_{12}$ can be obtained according to the Antoine equation. Then, the vapors of binary or ternary hydrocarbon mixtures with known liquid compositions were injected and measured on the GC. The saturated pressure of $C_6H_6$, $C_6H_{10}$, and $C_6H_{12}$:

$$\log P = A - \frac{B}{t + C} \qquad (3)$$

Here, $P$ is the pressure expressed in mmHg. A, B and C are constants (see Supplementary Table 2), and $t$ is the temperature expressed in °C.

The breakthrough vapor amount ($q_i$) was then calculated by integrating the flow rate $f(t)$ as following equation:

$$q_i = \frac{\int_0^{t_0} f(t)dt}{m} \qquad (4)$$

where the $m$ represents the mass used for the test.

The purity ($c$) of breakthrough vapor was then calculated by the following equation:

$$c = \frac{q_i}{q_i + q_i'} \qquad (5)$$

## Control experiments

The control experiments were conducted on ZSM-5, a benchmark adsorbent. Temperature-dependent adsorption tests for C6 cyclic hydrocarbons ($C_6H_6$, $C_6H_{10}$, $C_6H_{12}$) were performed on a TGA-Q50 gravimetric adsorption analyzer at three temperatures (30, 60, and 90 °C), following the same method and identical conditions as those employed in the case of Mn(DHBQ). Additionally, the separation performance of ZSM-5 on C6-ring mixtures was assessed by multi-component column breakthrough experiments at designated temperatures (30, 60, and 90 °C) under the identical conditions as for Mn(DHBQ).

## Heat flow measurements

The heat flows in the adsorption process were measured for $C_6H_6$, $C_6H_{10}$, and $C_6H_{12}$ by differential scanning calorimetry (DSC) on STD 650 (TA Instruments, Inc.). Runs at the thermobalance were performed by feeding a flow stream of C6 cyclic hydrocarbon diluted with nitrogen onto about 10 mg of evacuated sample at the designated temperature (30, 60 and 90 °C). Prior to measurements, the baseline was monitored under dry nitrogen flow at the same temperature (30, 60, and 90 °C), and then a nitrogen gas flow was introduced by bubbling the carrier gas in a saturator containing single-component liquid C6 cyclic hydrocarbon at a given temperature and the DSC signals were recorded. The scanning calorimetry curves were collected through real-time detection on the heat changes of the overall system inside the chamber during the adsorption process. The enthalpies (ΔH) were determined by integrating the heat change with respect to time.

## Ex situ powder X-ray diffraction (ex situ PXRD)

Ex situ PXRD coupled with vapor adsorption measurements were performed on a Rigaku Ultimate-IV diffraction system and equipped with thermogravimetric analyzer for hydrocarbon adsorption. Typically, about 20 mg as-made Mn-DHBQ was activated at 150 °C and cooled down to the designed temperature (30, 60, and 90 °C). The flow of nitrogen passed through the hydrocarbon bubbler to introduce it into the adsorption chamber. After achieving the adsorption equilibrium, the real-time Mn-DHBQ sample with vapor adsorption was quickly sent to PXRD system for the subsequent diffraction data collection.

## In situ infrared spectroscopy (in situ IR)

In situ IR coupled with vapor adsorption measurements were performed on a Nicolet 6700 infrared spectrometer equipped with a liquid nitrogen-cooled mercury cadmium telluride MCT-A detector and a temperature controller. Typically, the Mn-DHBQ samples (2-3 mg) were pressed onto a KBr pellet and placed into a vacuum cell placed at the focal point of the sample compartment of the infrared spectrometer. The cell was connected to a vacuum line for evacuation. All spectra were recorder under vacuum (base pressure <20 mTorr) in transmission mode with a frequency range of 600-4000 cm$^{-1}$ (4 cm$^{-1}$ spectral resolution). Before measurements, samples were activated at 150 °C and cooled down to the designed temperature (24, 60, and 90 °C). The vapor guest was vacuumed and introduced into the chamber for the stimuli-responsive sorption.

## Simulated computation

Fully non-local vdW-DF calculations were performed with the VASP code, using the standard projector augmented wave (PAW) pseudopotentials with a kinetic energy cutoff of 600 eV. Since the crystal structure for this Mn-DHBQ is consisting of periodic 1D chains, which are not chemically connected to each other but rather the interchain interaction is dominated by van der Waals forces, we modelled 1D chains with periodicity along the $a$ and $c$ crystallographic axes. Due to the large size of our unit-cell, only the Γ-point was sampled. In addition to the relaxation of the atomic positions, the periodic cell coordinates were also allowed to relax to mimic the flexibility of Mn-DHBQ. A strict criterion for geometry relaxation is adopted with an energy and forces convergence of $10^{-6}$ eV and 0.005 eV/Å, respectively. Binding energies were calculated as a difference of the individual and total energies:

$$E_b = E_{guest} + E_{MOF} - E_{guest+MOF} \qquad (6)$$

## Data availability

The processed and other derived data generated in this study are available in the Source Data file (https://doi.org/10.6084/m9.figshare.24846498). Source data are provided with this paper.

## Code availability

The codes and simulation files that support the plots and data analysis within this paper are available from the corresponding author upon request.

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

## Acknowledgements
We would like to thank the U.S. Department of Energy, Office of Science, Office of Basic Energy Sciences for supporting this work under grant No. DE-SC0019902. L. Chen and Z. Bao acknowledge the National Natural Science Foundation of China (No. 22225802, No. 22288102, and No. 22141001) for the partial support of this study. Computations were performed using the Wake Forest University High Performance Computing Facility, a centrally managed computational resource with support provided in part by the University.

## Author contributions
J. Li and F. Xie conceived the research idea and formulated the whole project. F. Xie synthesized the compounds, performed most of the physical characterizations, and collected the vapor adsorption data. Z. Bao, L. Chen, and Y. Fu carried out the vapor breakthrough and liquid separation experiments. K. Tan and E. Cedeño-Morales collected and analyzed the in situ IR data. T. Thonhauser and S. Ullah performed the theoretical calculations. J. Li and F. Xie coordinated the writing of the manuscript. All authors contributed to the drafting and revising of the manuscript and gave their approval to the final version of the paper.

## Competing interests
The authors declare no competing interests.
