## [Peer Review File · Nature Communications]

REVIEWER COMMENTS

Reviewer #1 (Remarks to the Author):

In this manuscript, the authors report a flexible one-dimensional coordination polymer ($\text{Mn}(\text{DHBQ})(\text{H}_2\text{O})_2$) to discriminate ternary C6 cyclic hydrocarbons (benzene, cyclohexene, cyclohexane) via a molecular sieving mechanism. This material shows high uptake ratios and selectivities for C6H6/C6H12 and C6H10/C6H12 mixtures. The separation mechanism was revealed by the combination of ex-situ PXRD analysis, in-situ IR spectroscopy study, and ab initio calculations. Overall, this appears interesting and complete study. However, the reviewer's concern is that the same material was already reported for p-, m-, and o-xylene isomers separation in *Science* (Li et al., *Science* 2022, 377, 335–339 (2022)) by the same group via a similar mechanism. In addition, the concept of temperature-dependent adsorption behavior illustrated in Scheme 1 was also reported in other flexible porous materials (*Nat. Chem.* 13, 933–939 (2021); *Angew. Chem. Int. Ed.* 2020, 59, 22756–22762). Therefore, this work lacks novelty enough for publication in nature communications.

Reviewer #2 (Remarks to the Author):

This manuscript presents the synthesis, structure, and adsorption/separation capabilities of C6 hydrocarbons using a novel porous coordination network formed from Mn(II) and DHBQ ligand. The primary 1D chains interconnect through hydrogen bonding, creating a 3D network with nanospaces among the chains. It's noteworthy that separating benzene, cyclohexene, and cyclohexane is a challenging task, yet critically important in the industrial sector. The material showcased in this study exhibits superior performance in separating these C6 hydrocarbons. Fascinatingly, its separation properties are temperature-dependent, allowing for refined purification of these molecules. The practical separation capabilities were validated through various methods, including single adsorption and mixed gas breakthrough experiments. The functional mechanisms were extensively unveiled through ex situ PXRD, in-situ IR, and theoretical calculations. The findings are both scientifically robust and intriguing, making them highly relevant to the wide readership of this journal. I would recommend for the publication of this manuscript in *Nature Communications*. I would hope the subsequent comments will be beneficial for the refinement of this research.

-Figure 1

It would be better to add the size of channel in the figure to understand the structure.

-(kinetic diameters: 5.85-6.0 Å),

Please add the reference of these data. In addition, the molecular dimensions would be better instead of kinetic diameters because benzene is not spherical but cyclohexane is closer to a sphere shape.

- single-component dynamic adsorption

More detailed measurements, such as flow rate, would be required to be documented, as the kinetic effect might be reflected in the experiment's results.

-The results from single-component adsorption measurements suggest that separation of the three C6 cyclic hydrocarbons by molecular sieving mechanism may be achieved

Please reconsider to use "molecular sieving". As my understanding, the molecular sieve should be used for the "Molecular size exclusion type separation". However, this case, the interaction between host and guest is more important. This is just a minor comment.

-benzene-cyclohexene-cyclohexane / C6H6/C6H10/C6H12

Please unify the terms as is possible.

-in situ infrared spectroscopy,

The section on in-situ IR is very verbose and hard to read. I recommend focusing on the essential parts in the main text and moving the rest to the Supporting Information.

-The upward (blue-) shift of $\sim 18 \text{ cm}^{-1}$ indicates the hardening and/or shortening of the C=C bond that agrees well with the observation of lattice constant decrease as identified by PXRD

I believe the shortening of the C=C bond isn't the primary cause for the decrease in the lattice constant. This is because the change in the length of the C=C bond should be much smaller than the change in the lattice. Please provide the quantitative values for the changes in both the C=C bond and the lattice constant?

-Discussion section,

The Discussion Section currently contains only a summary of the paper. Please reconsider either the section heading or the content within the discussion.

Reviewer #3 (Remarks to the Author):

This is an interesting manuscript focusing on the use of coordination polymers for vapor-phase C6 separations. The authors suggest that the flexibility of the polymer may be the key role in the separation. The team highlights experimental and modeling work that interrogates the C6 interactions with the open metal sites within the coordination polymer.

The paper has potential, but there are some issues preventing recommendation of acceptance at this stage.

1. There have been many papers focusing on MOF or ZIF adsorbents separating small molecules based on conformational changes in the MOF or ZIF structure (e.g., Kapteijn and the ZIF-7 ethylene/ethane work). However, one puzzle has never really been explained in a satisfactory way: if one of the molecules in the mixture induces a structural change in the coordination polymer, won't all the of the molecules now be able to interact with the new new structure?

This question seems pertinent for this work - it would be great if the authors can provide some comments on this conundrum.

2. A more serious issue is that the authors attribute the temperature-induced changes to flexibility of the polymer. A simpler alternative that the authors should rule out via calculations is that the C6 molecules simply have different binding energies, and thus their isotherms will have differing temperature dependencies. The authors should show with calculations (perhaps the van't Hoff approach) that the changes in the isotherm do not follow standard adsorption thermodynamics models.

3. The final major issue with the article is that there does not seem to be any efforts towards reproducibility or even control experiments. This is not up to the standards of Nature Communications. The authors are encouraged to resynthesize and retest (and describe these) to build confidence in the results. The fact that the words "error" "deviation" "reproducibility" do not show up in the manuscript is a red flag. Moreover, the authors should include a control material that every lab can test to demonstrate the enhancements from the new material.

RESPONSE TO REVIEWERS:

We are grateful to the reviewers for their generally very positive remarks, valuable comments, and constructive suggestions. We have carried out a series of additional experimental and theoretical work in the revision process and included new data and analysis in our revised documents. We have carefully considered and answered each and every question raised by all three reviewers and revised our manuscript and supplementary information accordingly. All changes and additions are marked in RED colored text.

Reviewer #1:

In this manuscript, the authors report a flexible one-dimensional coordination polymer (Mn(DHBQ)(H₂O)₂) to discriminate ternary C₆ cyclic hydrocarbons (benzene, cyclohexene, cyclohexane) via a molecular sieving mechanism. This material shows high uptake ratios and selectivities for C₆H₆/C₆H₁₂ and C₆H₁₀/C₆H₁₂ mixtures. The separation mechanism was revealed by the combination of ex-situ PXRD analysis, in-situ IR spectroscopy study, and ab initio calculations. Overall, this appears interesting and complete study. However, the reviewer's concern is that the same material was already reported for p-, m-, and o-xylene isomers separation in Science (Li et al., Science 2022, 377, 335–339 (2022)) by the same group via a similar mechanism. In addition, the concept of temperature-dependent adsorption behavior illustrated in Scheme 1 was also reported in other flexible porous materials (Nat. Chem. 13, 933–939 (2021); Angew. Chem. Int. Ed. 2020, 59, 22756–22762). Therefore, this work lacks novelty enough for publication in nature communications.

Response: The reviewer has a generally positive opinion on our manuscript: "Overall, this appears interesting and complete study". The main concern expressed by this reviewer is that "...the same material was already reported for p-, m-, and o-xylene isomers separation in Science (Li et al., Science 2022, 377, 335–339 (2022)) by the same group via a similar mechanism". Respectfully, while the material has been previously reported for effective separation of xylene isomers, the C₆-ring molecules investigated in the current work are a group of hydrocarbons totally different from C₈ alkylbenzene isomers, structurally and chemically. A given adsorbent may or may not have the same capability for effective separation of different types of molecules, which will need to be determined by a careful and thorough study. In addition, the two papers cited by the reviewer are on the separation of ethylene and ethane (C₂ hydrocarbons) by a hydrogen-bonded organic framework (Nat. Chem. 13, 933–939, 2021) and separation of a ternary gas mixture C₂H₂/CO₂/C₂H₄ (Angew. Chem. Int. Ed. 2020, 59, 22756–22762). Both papers deal with entirely different adsorbate and adsorbent systems.

Reviewer #2:

This manuscript presents the synthesis, structure, and adsorption/separation capabilities of C₆ hydrocarbons using a novel porous coordination network formed from Mn(II) and DHBQ ligand. The primary 1D chains interconnect through hydrogen bonding, creating a 3D network with nanospaces among the chains. It's noteworthy that separating benzene, cyclohexene, and cyclohexane is a challenging task, yet critically important in the industrial sector. The material showcased in this study exhibits superior performance in separating these C₆ hydrocarbons. Fascinatingly, its separation properties are temperature-dependent, allowing for refined

purification of these molecules. The practical separation capabilities were validated through various methods, including single adsorption and mixed gas breakthrough experiments. The functional mechanisms were extensively unveiled through ex situ PXRD, in-situ IR, and theoretical calculations. The findings are both scientifically robust and intriguing, making them highly relevant to the wide readership of this journal. I would recommend for the publication of this manuscript in Nature Communications. I would hope the subsequent comments will be beneficial for the refinement of this research.

Q1. -Figure 1: It would be better to add the size of channel in the figure to understand the structure.

Response: We thank the reviewer for the suggestion. We have added estimated channel size ($\sim 5.5 \text{ \AA}$) in Figure 1e and a short description in the figure caption.

Q2. -(kinetic diameters: 5.85-6.0 \AA): Please add the reference of these data. In addition, the molecular dimensions would be better instead of kinetic diameters because benzene is not spherical but cyclohexane is closer to a sphere shape.

Response: Very good point. Following the reviewer's suggestion, we have added molecular dimensions for all three C6 cyclic hydrocarbons in the revised manuscript (page 6). We have also added references for the kinetic diameters (Table S1).

Q3. - single-component dynamic adsorption: More detailed measurements, such as flow rate, would be required to be documented, as the kinetic effect might be reflected in the experiment's results.

Response: We thank the reviewer for this helpful comment. We have added more details (including flow rate) of our single-component dynamic adsorption experiments (see revised text, page 19). We have also included more details in the discussion and comparison of adsorption kinetics of three C6 cyclic hydrocarbons (see revised text, page 7 and Figures S16-S18).

Q4. -The results from single-component adsorption measurements suggest that separation of the three C6 cyclic hydrocarbons by molecular sieving mechanism may be achieved. Please reconsider to use "molecular sieving". As my understanding, the molecular sieve should be used for the "Molecular size exclusion type separation". However, this case, the interaction between host and guest is more important. This is just a minor comment.

Response: We thank the reviewer for raising this point and we agree that interaction between host and guest plays the most important role in the adsorption of three C6-ring hydrocarbons. Indeed, the term "molecular sieving" conventionally refers to the size-exclusion based separation of molecules due to the differences in their sizes. Molecules having their physical dimensions smaller than the pore size can be adsorbed while those having their dimensions larger than the pore size will be excluded. For rigid porous materials, the pore size remains essentially unchanged upon gas/vapor adsorption. In the case of 1D-Mn(DHBQ), however, the pore structure is highly "flexible". The pore size (or interchain space) of its activated structure is too small to fit in any of the three C6 molecules (as evident from the PXRD analysis). However, change (or more precisely, expansion) of the pore (interchain space) may occur depending on the temperature and individual adsorbate molecule. At a given temperature, if a particular molecule (A) is capable of stretching/expanding the pore space sufficiently large

it will be adsorbed. At the same temperature, another molecule (B) may be totally excluded (sieved out) if it can't stretch the pore open sufficiently large to allow it to enter. However, the same molecule (B) may be able to expand the pore space at a lower temperature and get adsorbed at large quantity. Accordingly, the separation mechanism will be different for the same molecule at different temperatures. This behavior is consistent with the heat released (enthalpy) during the adsorption process measured by DSC, which varies significantly as a function of temperature (see revised text, page 8 and Figure 2c-2e). Based on these observations and findings, we hope the expression "temperature-dependent molecular sieving" is proper to reflect the unique adsorption behavior and complex separation mechanism associated with the structure change at different temperatures.

Q5. -benzene-cyclohexene-cyclohexane / C₆H₆/C₆H₁₀/C₆H₁₂ Please unify the terms as is possible.

Response: Done.

Q6. -in situ infrared spectroscopy: The section on in-situ IR is very verbose and hard to read. I recommend focusing on the essential parts in the main text and moving the rest to the Supporting Information.

Response: As suggested by the reviewer, we have moved the entire section of the assignment of sample phonon modes to Supplementary Note S1 (see Supplementary Information, page 84). We have also substantially shortened the description on sample activation step in the revised text (page 12).

Q7. -The upward (blue-) shift of $\sim 18\text{ cm}^{-1}$ indicates the hardening and/or shortening of the C=C bond that agrees well with the observation of lattice constant decrease as identified by PXRD. I believe the shortening of the C=C bond isn't the primary cause for the decrease in the lattice constant. This is because the change in the length of the C=C bond should be much smaller than the change in the lattice. Please provide the quantitative values for the changes in both the C=C bond and the lattice constant?

Response: We thank the reviewer for careful reading of our manuscript and apologize for not making our description more clear. Actually, we haven't claimed that the shortening of the C=C bond (as indicated by red-shift of $\nu(\text{C}=\text{C})$ mode) is the primary cause for the decrease in the lattice constant. We fully agree with the reviewer that the changes in the lattice constant is at a much larger extent than that of the C=C bond. While infrared (IR) spectroscopy serves as a sensitive probe to reveal the local changes occurring to the specific bond, e.g., bond shorten/elongating, usually accompanied by blue or red-shift of its fundamental stretch mode (see *Power of Infrared and Raman Spectroscopies to Characterize Metal-Organic Frameworks and Investigate Their Interaction with Guest Molecules. Chemical Reviews 2021. 121,1286-1424*), the quantitative determination of bond length is still not possible through IR. Thus, we unfortunately could not provide quantitative value for the change in the C=C bond.

Q8. -Discussion section, The Discussion Section currently contains only a summary of the paper. Please reconsider either the section heading or the content within the discussion.

Response: We appreciate the reviewer's valuable suggestion. We have integrated the discussion section with the results section in the revised manuscript. This modification provides a more cohesive and succinct presentation of both the findings and interpretations.

Reviewer #3:

This is an interesting manuscript focusing on the use of coordination polymers for vapor-phase C6 separations. The authors suggest that the flexibility of the polymer may be the key role in the separation. The team highlights experimental and modeling work that interrogates the C6 interactions with the open metal sites within the coordination polymer.

The paper has potential, but there are some issues preventing recommendation of acceptance at this stage.

Q1. There have been many papers focusing on MOF or ZIF adsorbents separating small molecules based on conformational changes in the MOF or ZIF structure (e.g., Kapteijn and the ZIF-7 ethylene/ethane work). However, one puzzle has never really been explained in a satisfactory way: if one of the molecules in the mixture induces a structural change in the coordination polymer, won't all the of the molecules now be able to interact with the new new structure? This question seems pertinent for this work - it would be great if the authors can provide some comments on this conundrum.

Response: We thank the reviewer for this legitimate and very important question. We will answer this question based on the results of our experiments on single-component and equimolar mixture systems of the C6-cyclic hydrocarbon molecules at temperatures where some of these molecules would be excluded (sieved out) under pure phase (single-component) adsorption because they are incapable to expand the pore space (open the gate) at these temperatures. For example, at 90 °C, pure benzene can be adsorbed with high loading but cyclohexane is totally excluded in their single-component adsorption experiments (Figure S11). From the data obtained in equimolar mixture at the same temperature (90 °C), both molecules show adsorption (see NMR spectra in Figure S64), although with a much higher amount of benzene (adsorption ratio of benzene to cyclohexane: 108 to 1). These results show that cyclohexane can indeed enter the channel upon pore expansion by benzene) but with a much lower adsorption capacity. This is because having a significantly higher binding energy (0.57 eV) and adsorption enthalpy (43.6 kJ mol⁻¹) at 90 °C and faster kinetics, benzene interacts much more strongly with the adsorbent, thus taking up most binding sites much more quickly than cyclohexane in competitive adsorption of an adsorbate mixture.

Q2. A more serious issue is that the authors attribute the temperature-induced changes to flexibility of the polymer. A simpler alternative that the authors should rule out via calculations is that the C6 molecules simply have different binding energies, and thus their isotherms will have differing temperature dependencies. The authors should show with calculations (perhaps the van't Hoff approach) that the changes in the isotherm do not follow standard adsorption thermodynamics models.

Response: The binding energies at the *ab initio* level (already reported in the original manuscript) but we have meanwhile also calculated the *ab initio* temperature-dependent

binding enthalpies as a function of temperature (Figure S74b). Integrating the Van't Hoff equation from 60 to 90 °C, we find that the thermodynamic equilibrium constant changes by a factor of 4.8, 3.7, and 3.2 for C₆H₆, C₆H₁₀, and C₆H₁₂ when going from 60 to 90 °C. Using a simple Langmuir isotherm model, we can estimate the expected temperature change in the isotherms if we assume that adsorption follows a standard adsorption thermodynamics model. Our predictions under this assumption are shown in Figure A below. Compared to the experimental data at the same temperatures, it becomes clear that they deviate significantly from the standard adsorption thermodynamics prediction, and hence, non-standard adsorption thermodynamics is at play.

Figure A. Predicted temperature dependence of the adsorption isotherms, assuming standard adsorption thermodynamics and using a simple Langmuir isotherm model. The temperature dependence was obtained by integrating Van't Hoff's equation. C₆H₆ (red), C₆H₁₀ (blue), and C₆H₁₂ (green). Solid and dotted lines are for data at 60 and 90 °C, respectively.

The non-standard adsorption thermodynamics of the C₆-cyclic molecules is closely related to the structural flexibility and temperature. The structure change (pore expansion) required for a particular molecule to be adsorbed will occur only when sufficient energy is provided to overcome the strain. To illustrate this, we have also measured heat change (ΔH , enthalpy) during the adsorption of each C₆ molecule at all three temperatures (30, 60 and 90 °C) by differential scanning calorimetry (DSC). As depicted in Figure 2c-2e and briefly discussed in the revised manuscript (page 8), the values are very different for the three molecules at different temperatures. Taking C₆H₁₂ as an example, the low values of 13.0 and 11.5 kJ mol⁻¹ at 60 and 90 °C are simply too low to offer sufficient energy to open the pore space and thus it is sieved out at these temperatures. This behavior is different from that at lower temperature (e.g., 30 °C), where sufficient energy ($\Delta H = 27.9$ kJ mol⁻¹) enables pore expansion and standard adsorption thermodynamics may apply. Similar observations are found for the other two C₆ molecules.

Q3. The final major issue with the article is that there does not seem to be any efforts towards reproducibility or even control experiments. This is not up to the standards of Nature Communications. The authors are encouraged to resynthesize and retest (and describe these) to build confidence in the results. The fact that the words "error" "deviation" "reproducibility" do

not show up in the manuscript is a red flag. Moreover, the authors should include a control material that every lab can test to demonstrate the enhancements from the new material.

Response: We thank the reviewer for this valuable suggestion, and we fully agree that reproducibility tests and control experiments are important. In response to these concerns, we have conducted a series of experiments to assess the reproducibility of Mn(DHBQ) (see revised text, pages 22 and 24, for experimental details), and briefly discussed the results in the revised manuscript (pages 9-10) and supplementary information (Figures S48-S51). We prepared six batches of Mn-DHBQ samples using the same method. Quantitative yields (>95%) were achieved for all six batches of samples (Figure S48). Structural characterizations by PXRD analysis (Figure S49) demonstrate excellent reproducibility in terms of crystallinity and quality. Porosity assessment (Figure S50), including BET surface area and pore size distribution estimated from N₂ isotherm data measured at 77 K, further confirms the excellent reproducibility across multiple batches. Furthermore, we have measured temperature-dependent adsorption isotherms for each of the six samples (Figure S51). The results confirm the experiments are fully reproducible.

For control experiments, we have selected commercially available, benchmark adsorbent ZSM-5 as a reference material. The adsorption and separation experiments were performed at the same conditions as used for Mn(DHBQ). The results are depicted in Figures S52-S54 in the revised SI. These results highlight the performance of Mn(DHBQ) is significantly better than ZSM-5. Specifically, the uptake amounts of C₆H₆ are much lower for ZSM-5 (1.15, 0.7 and 0.6 mmol/g) compared to Mn-DHBQ (2.55, 2.32 and 2.23 mmol/g) at 30, 60 and 90 °C, respectively. Moreover, the separation performance of ZSM-5 is noticeably inferior to Mn(DHBQ), with very short breakout times for each C₆ molecule at 30 and 60 °C and no discernible separation at 90 °C (all breaking out at the same time).

Finally, We hope that these changes and revisions are satisfactory to meet all reviewers' expectations, and we extend our gratitude once more to all the reviewers for their invaluable insights, comments, and suggestions. Their constructive feedback has undoubtedly played a crucial role in enhancing the overall quality of this manuscript.

REVIEWER COMMENTS

Reviewer #2 (Remarks to the Author):

The manuscript has been thoroughly revised, and my previous concerns have been fully addressed. I would recommend the publication of this paper in Nature Communications as it stands.

Reviewer #3 (Remarks to the Author):

I appreciate the authors' careful responses and revision to the manuscript. In general, I find the manuscript is technically correct, but two changes should occur prior to recommendation of acceptance:

1. The authors should more seriously respond to Reviewer 1's novelty concerns by updating the manuscript text.

2. Figure A in the response letter should be added to the SI and discussed briefly somewhere in the manuscript. I agree that the isotherms the authors show are deviating from general adsorption thermodynamics, but perhaps not by as much as the authors are suggesting. A somewhat more quantified version of this comparison (Figure A vs. actual isotherms) should be added.

RESPONSE TO REVIEWERS:

Reviewer #2 (Remarks to the Author):

The manuscript has been thoroughly revised, and my previous concerns have been fully addressed. I would recommend the publication of this paper in Nature Communications as it stands.

Response: We are very pleased to know that the reviewer is totally satisfied with the revision.

Reviewer #3 (Remarks to the Author):

I appreciate the authors' careful responses and revision to the manuscript. In general, I find the manuscript is technically correct, but two changes should occur prior to recommendation of acceptance:

1. The authors should more seriously respond to Reviewer 1's novelty concerns by updating the manuscript text.

Response: We have further emphasized the novelty and significance of this work in the revised text (see page 3). Briefly, very few studies have been reported to this date on adsorptive separation of C6 cyclic hydrocarbons. This work not only represents the only study of adsorption-based separation of benzene-cyclohexene-cyclohexane ternary mixtures but also discloses a unique example of complete separation of the C6 mixtures by temperature-dependent molecular sieving mechanism with very high selectivity. We also hope the current study inspires future efforts of the MOF community in exploring the very interesting temperature-dependent adsorption properties and gaining in-depth understanding of the structure-adsorption property correlations in flexible adsorbent materials.

2. Figure A in the response letter should be added to the SI and discussed briefly somewhere in the manuscript. I agree that the isotherms the authors show are deviating from general adsorption thermodynamics, but perhaps not by as much as the authors are suggesting. A somewhat more quantified version of this comparison (Figure A vs. actual isotherms) should be added.

Response: Following the reviewer's suggestion, we have calculated temperature dependence of isotherms using a simple Langmuir isotherm model and compared them with experimentally measured data at all three temperatures (30, 60 and 90 °C). The results are plotted in Figures S76-S78.

Based on these data, it is clear the experimental isotherms deviate significantly from what standard thermodynamics would predict, especially at higher temperatures. A short summary has also been added to the revised text (see page 12).

We hope that these changes are satisfactory, and we would like to thank Reviewer 3 again for the additional comments and suggestions, which have helped to further improve the quality of this manuscript.

REVIEWERS' COMMENTS

Reviewer #3 (Remarks to the Author):

The article has been sufficiently revised and I recommend it for publication.